# The Ecology of the Zebra Finch Makes It a Great Laboratory Model but an Outlier amongst Passerine Birds

**Simon C. Griffith ***, **Riccardo Ton, Laura L. Hurley**, **Callum S. McDiarmid and Hector Pacheco-Fuentes**

Department of Biological Sciences, Macquarie University, Sydney, NSW 2109, Australia;
riccardo.ton@mq.edu.au (R.T.); laura.hurley@mq.edu.au (L.L.H.); callum.mcdiarmid@hdr.mq.edu.au (C.S.M.);
hector.pacheco@hdr.mq.edu.au (H.P.-F.)
**\*** Correspondence: simon.griffith@mq.edu.au

**Simple Summary:** The Zebra Finch (*Taeniopygia guttata*) is the focus of more work in the laboratory than any other non-production bird. It has been so widely used because of key physiological, behavioural and life history characteristics. The species can be maintained and bred easily on a very poor diet and is the fastest maturing bird in the world. Here, we argue that whilst these characteristics do indeed make it an excellent species for conducting captive experiments in a controlled environment, they also make it an outlier amongst birds. Therefore, in many research fields, and particularly those focused on behaviour and life history evolution, great care needs to be taken in using the species appropriately and interpreting the results. The Zebra Finch is unlikely to be a good 'model' of general bird life history or behaviour, and is certainly very different from many of the well-studied birds in the northern hemisphere. Our paper should help to direct future research on the Zebra Finch in the wild and laboratory.

**Abstract:** Zebra Finches have become the most widely researched bird species outside of those used in agricultural production. Their adoption as the avian model of choice is largely down to a number of characteristics that make them easy to obtain and use in captivity. The main point of our paper is that the very characteristics that make the Zebra Finch a highly amenable laboratory model species mean that it is by definition different from many other passerine birds, and therefore not a good general model for many research areas. The Zebra Finch is likely to be particularly resilient to the effects of stress early in life, and is likely to show great flexibility in dealing with a wide variety of conditions later in life. Whilst it is tempting for researchers to turn to species such as the Zebra Finch, that can be the focus of manipulative work in the laboratory, we caution that the findings of such studies may confound our understanding of general avian biology. The Zebra Finch will remain an excellent species for laboratory work, and our paper should help to direct and interpret future work in the laboratory and the field.

**Keywords:** model system; *Taeniopygia guttata*; evolutionary ecology; early developmental effects; life history; dietary manipulation



## 1. Introduction

A species becomes a model in science when it combines practical properties that facilitate its easy maintenance, rearing and manipulation with characteristics that make it suitable for addressing particular research questions [1,2]. Although not one of the big six model species that have been so influential in modern biology through extensive laboratory research [1], the Zebra Finch is one of the most researched avian species over the past few decades (see Table 1). The Zebra Finch compares with other classic model systems such as *Caenorhabditis elegans*, *Drosophila melanogaster*, and *Mus musculus* [1,3,4] in that it adjusts well to captivity, has a relatively high annual fecundity, early sexual maturity, a simple diet (dry seed) [5], and the tendency to breed under a broad spectrum

of environmental conditions in the laboratory [6]. This makes the Zebra Finch a very convenient species for those researchers wishing to pursue research questions focused on a bird in the standardized conditions of the laboratory. Another 'convenience' is that the Zebra Finch is one of the most commonly and widely kept companion birds globally and is readily available from pet stores around the world. Most Zebra Finch laboratory research populations have been sourced from these avicultural populations accessed through pet shop stock [7]. This is an interesting departure from the classic model systems that are available as genetically standardized commercially available strains [8]. Even in the absence of highly standardized genetic strains, there is little doubt that the Zebra Finch is an extremely good and appropriate species to examine certain areas of science such as neurobiology.

**Table 1.** The number of papers found on searches for a collection of the most widely researched avian species (this is an indicative analysis and there may be some well-researched species that were not searched). From Web of Science (Clarivate Analytics), accessed on the topic search "[scientific name of each species]" on 18 December 2020 searching for "Topic" between 2000 and 2020. "Topic" searches title, abstract, author keywords and keywords plus.

| English Name | Scientific Name | Papers 2000–2020 |
|---|---|---|
| Feral/Domestic Pigeon | *Columba livia* | 4177 |
| Great Tit | *Parus major* | 4035 |
| Quail/Japanese Quail | *Coturnix coturnix japonica* | 3639 |
| House Sparrow | *Passer domesticus* | 2940 |
| Zebra Finch | *Taeniopygia guttata* | 2937 |
| Common Starling | *Sturnus vulgaris* | 2608 |
| Blue Tit | *Cyanistes caeruleus* * | 1970 |
| Barn Swallow | *Hirundo rustica* | 1606 |
| Pied Flycatcher | *Ficedula hypoleuca* | 1354 |
| Budgerigar | *Melopsittacus undulatus* | 908 |
| Song Sparrow | *Melospiza melodia* | 889 |
| Chaffinch | *Fringilla coelebs* | 833 |
| Tree Swallow | *Tachycineta bicolor* | 822 |
| Collared Flycatcher | *Ficedula albicollis* | 646 |
| Dark-Eyed Junco | *Junco hyemalis* | 587 |
| Red-Winged Blackbird | *Agelaius phoeniceus* | 546 |
| House Finch | *Carpodacus mexicanus* | 534 |
| Canary | *Serinus canaria* | 529 |
| Red-Legged Partridge | *Alectoris rufa* | 527 |
| Cockatiel | *Nymphicus hollandicus* | 398 |
| Indian Peafowl | *Pavo cristatus* | 378 |

* Due to a change in the name of the genus, the searches for *Parus caeruleus* and *Cyanistes caeruleus* were summed together for Blue Tit.

The Zebra Finch has become established as the main model species for understanding the neuronal mechanisms of song and vocal learning [9,10], and more recently the female perception of song [11]. The utility of the Zebra Finch in this research field followed the early discovery of the clearly defined sensitive period in development for the acquisition of phenotypes relating to acoustic production and perception [12]. However, the establishment of the Zebra Finch as the model of choice for this research field of neurobiology and song learning was driven by the characteristics that the Zebra Finch shares in common with the classic biological model systems—they are easy and relatively inexpensive to keep and rear in captivity, and they have a quick generational turnover [1,4]. Over the past twenty years, just over half of the Zebra Finch research publications have been in the field of neurobiology (1542 out of 2937 papers—Web of Science analysis conducted on 18 December 2020; summarised in Table 2).

**Table 2.** The use of the Zebra Finch in the primary literature from 2000 to 2020. Non-mutually exclusive research fields from Web of Science (Clarivate Analytics), accessed on the topic search "*Taeniopygia guttata*" on 18 December 2020.

| Research Areas | Records | % of 2937 |
|---|---|---|
| Behavioural sciences | 1867 | 64 |
| Neurosciences neurology | 1542 | 53 |
| Biochemistry molecular biology | 1258 | 43 |
| Reproductive biology | 1028 | 35 |
| Genetics heredity | 984 | 34 |
| Environmental sciences ecology | 913 | 31 |
| Physiology | 847 | 29 |
| Psychology | 781 | 27 |
| Evolutionary biology | 661 | 23 |
| Endocrinology metabolism | 601 | 20 |
| Developmental biology | 582 | 20 |
| Anatomy morphology | 465 | 16 |
| Nutrition dietetics | 394 | 13 |
| Immunology | 347 | 12 |
| Pharmacology pharmacy | 334 | 11 |
| Communication | 322 | 11 |
| Cell biology | 278 | 9 |
| Toxicology | 148 | 5 |

In this clearly defined research field, the Zebra Finch will continue to be successfully used as a model to understand the neuronal and genomic pathways through which an organism can acquire and express complex acoustic signals, that in turn can help with the understanding of language in humans [13]. The Zebra Finch is an appropriate model here because it is a well-defined field with clear boundaries in which researchers are focused primarily in explaining how things work mechanistically in this species as a model of how such phenomena can operate in an animal. The 'representational target' of the research conducted on the neurobiology of song learning in the Zebra Finch in this field is relatively narrow, and the 'representational scope' may also be limited, although the insight delivered by this research program may be very significant, as we understand how complex brains work in this context (see [14] for further discussion of these epistemological issues). Other good examples of highly defined mechanistic questions in which the Zebra Finch has been adopted as a suitable model include the study of ecotoxicology and immunology. For example, recent work in the laboratory has focused on how Mercury (Hg) affects a variety of traits in the Zebra Finch in an effort to understand avian responses in a general way to this common metal pollutant [15]. Similarly, in an effort to understand the likely effects of the emergent West Nile virus on birds generally, the Zebra Finch has been suggested as an appropriate passerine species in which laboratory trials can be conducted [16]. Even though there will be differences in the immune function of different laboratory populations [17], and indeed differences between wild and laboratory populations in the underlying architecture of immune components such the Major Histocompatibility Complex [18], the Zebra Finch will remain one of the best species in which to conduct basic research on such highly mechanistic questions.

In parallel with this use of the Zebra Finch as a model for understanding highly focused and mechanistic questions, the Zebra Finch has also been widely used in fields such as behavioural biology, reproductive biology, and physiology (Table 2). However, whilst there is the convenience of the Zebra Finch as a highly accessible laboratory species for research in these areas, the representational target is not so narrowly defined in these fields (as in the song-learning research), and that is likely to lead to problems. Indeed, the features that have made the species so well suited as a laboratory subject are likely to significantly constrain the representational scope of research outcomes, and at worst confound our understanding of certain fields of biology. The problem stems from the fact

that the key characteristics that make species good subjects for laboratory research [2] set them apart from other species in the taxa that they represent—they are outliers and, as a result, the conclusions that can be drawn from research on them might poorly reflect that branch of biodiversity. For example, it has been argued that the laboratory mouse differs from close relatives in key life history and behavioural traits that adapted them for success as a commensal species and also made them so useful as a laboratory model [19]. Similarly, Markow [20] has argued that *Drosophila melanogaster* is an extreme ecological generalist that became a human commensal species and rapidly expanded out of Africa over the past 15,000 years, adapting to a wide range of environmental conditions as they went [20]. In addition to the suite of characteristics that pre-adapted model species to cope with laboratory conditions, laboratory populations also differ further from their wild congeners through the selection exerted as they were shaped into a more useful laboratory model [8].

If a Zebra Finch does not represent an average passerine in key behavioural and life history traits, that is not a problem for those seeking to understand the parts of the brain which control the production of song. However, it will be a problem for those studying avian life history more broadly and attempting to use the Zebra Finch as an experimental model system to represent birds more generally. An additional issue is that unlike the laboratory mouse and fruit fly, the Zebra Finch is not a globally widespread human commensal species and, as a result, most of those conducting research on the species will not be familiar with its ecological context. As shown in Table 3, summarising the research effort on the Zebra Finch over the past 20 years, only 6% of research outputs have come from the only country, Australia, in which the focal subspecies is found in the wild. (Nearly all laboratory work is focused on the Australian subspecies, which diverged from the Timor subspecies a few million years ago [Zann 1996].) Again, it is not a problem for a neuroscientist in the US or the UK to work on a species with which they are have no familiarity in the wild, because their research field is highly mechanistic and requires no ecological context. It may, however, be a problem for those working in a behavioural field, who have no familiarity with the natural behaviour of the Zebra Finch, or indeed the natural environment in which they live.

**Table 3.** The origin of the published research focused on the Zebra Finch from 2000 to 2020. From Web of Science (Clarivate Analytics), accessed on the topic search "*Taeniopygia guttata*" on 18 December 2020.

| Countries/Regions | Records | % of 2937 |
|---|---|---|
| USA | 1431 | 49 |
| UK | 851 | 29 |
| Germany | 372 | 13 |
| Canada | 279 | 9 |
| Australia | 189 | 6 |
| Netherlands | 157 | 5 |
| France | 153 | 5 |
| China | 130 | 4 |
| Japan | 112 | 4 |
| Sweden | 96 | 3 |
| Belgium | 71 | 2 |
| Spain | 71 | 2 |
| New Zealand | 60 | 2 |
| Switzerland | 43 | 1 |
| Italy | 33 | 1 |

In contrast to the work on neuroscience, and other similarly mechanistic research fields, for the reasons outlined below, we believe it might be a problem for a biologist who works on life history, or behavioural questions in birds in the northern hemisphere, to use the Zebra Finch as a convenient laboratory species to address questions that are often

conceptually conceived through their familiarity with their local northern hemisphere birds. Indeed, anecdotally, it is noticeable that many researchers who conduct laboratory research on the Zebra Finch are often concurrently researching local species in the wild in Europe and North America. It may be difficult to bring those local wild species into the laboratory in order to conduct the kind of controlled experiments that can be achieved with the Zebra Finch. However, we argue that, despite being broadly perceived as similar, the Zebra Finch is a fundamentally different species from the northern hemisphere passerines that are widely studied in the wild. For a variety of reasons outlined below, the Zebra Finch is unlikely to be a good surrogate to model research questions that are conceived from wild studies of northern hemisphere, seasonally breeding passerines such as the Great Tit, Blue Tit, European Pied Flycatcher, Tree Swallow, House Finch, etc.

Indeed, whilst the Zebra Finch is more similar to many other Australian birds in aspects of its behaviour, physiology and ecology, even within the Australian avifuana, it is likely to be an outlier (although the extent to which this is true has yet to be examined). The Zebra Finch has evolved in the extreme arid zone that comprises over 75% of the Australian continent and is characterised by extreme heat, aridity and unpredictable cycles of rainfall that vary temporally, spatially and in their magnitude (reviewed in [21]). Australian birds are generally far more flexible in their breeding phenology than birds in the northern hemisphere temperate zone [22], and indeed the Zebra Finch is well recognised as a highly opportunistic breeder that is capable of breeding at any time of the year as long as food is available [5,23]. A recent analysis of flexible breeding phenology across over 300 species of terrestrial birds in Australia found that arid zone birds such as the Zebra Finch actually show a more constrained pattern of breeding than species in the less arid coastal fringe [22]. Nevertheless, temperate Australian species generally have breeding seasons that were found to be 2.3-fold longer than species of songbirds breeding in a temperate region of the northern hemisphere [22]. So, both Zebra Finches, and Australian birds generally are likely to be quite different to the wild birds of the same order that are local and familiar to most of those studying Zebra Finches in the laboratory (in the northern hemisphere). Indeed, nearly all of the research on the Zebra Finch over the past couple of decades has been conducted in a relatively small number of countries in the temperate northern hemisphere (approximately 80% of the work in the past two decades has been conducted in the USA, UK, Canada, Germany, Netherlands, Sweden, and France; see Table 3).

Here we highlight some areas in which recent work on the Zebra Finch has demonstrated that the species does not respond to parameters in the way that we might expect conceptually based on findings in other avian species. We have been involved in some of this work and we acknowledge some naivety in the conception of that work. This naivety is partly due to the fact that our understanding of avian behaviour and life history is, to a large extent, based on studies of a relatively small number of very well researched passerine species in the northern hemisphere. This paper is partly our response to that, and an effort to raise awareness to prevent future work by others repeating the mistake. Finally, we hope that the perspective that we provide on the Zebra Finch may improve the interpretation of Zebra Finch studies already in the literature. The areas that we present are all related to aspects of the life history variation of the species. We focus on the response of adult Zebra Finches to variation in diet and the effects of climate on development and reproduction, and then review how the overall life history of the Zebra Finch fits into existing paradigms.

### 1.1. Food Quality/Abundance, Reproductive Success and Development

In well-studied passerine species of the northern hemisphere such as the Great Tit, there is a demonstrated relationship between food abundance and reproductive investment and success (e.g., [24]). Seasonally breeding birds of the northern hemisphere are generally thought to be *capital* breeders, investing in reproduction in line with stored resources [25]. The Zebra Finch, living in an ecologically unpredictable environment with an extended breeding period [23], seems more likely to be an *income* breeder, tailoring its investment with

concurrently acquired resource availability [25]. However, in reality, these two investment strategies are best considered as two extremes of a continuum [26], and indeed some species switch between the two depending on ecological conditions [27]. It would be useful to identify the nutritional conditions that trigger reproduction and investment decisions in the Zebra Finch, and how the species fits in the paradigm of capital and income breeding.

Although experimentally the relationship between nutrition and reproduction is relatively poorly understood in the Zebra Finch (but see refs below), typically in surveyed laboratory populations (18 out of 19), researchers maintained and bred their Zebra Finches with a standard dry seed diet and high-quality supplements such as boiled (chicken) egg, or egg and biscuit supplements [6]. Clutch size varies across females and populations [6], and in the laboratory, clutch size, to some extent, reflects nutritional conditions either before, or during egg laying [28–31]. However, all of the studies cited directly above manipulated diet using supplements based on chicken eggs. Given that Zebra Finches do not normally consume any animal protein [5], this probably represents a super stimulus. Whilst the use of such egg-based dietary manipulations may have been appropriate in the context of those experimental studies [28–31], a question remains about the extent to which reproductive investment may vary in the context of a high- or low-quality plant-based diet. As described in the Supplementary Materials (SM1), we experimentally tested the effect of high- and low-quality plant-based diets on both domesticated and wild derived Zebra Finches. Surprisingly, we found no indication of any difference in the reproductive performance of pairs that were kept on just a mix of dry seeds, against similar pairs supplemented daily with a highly nutritious 'green mix' comprised of a broader range of dry seeds, green seed, vitamin and mineral mix, and fresh 'snap frozen' vegetables (carrot, broccoli, green beans, cauliflower) (full details in SM1). We investigated a range of reproductive traits and performance measures, such as clutch size, and the number of eggs that hatched and chicks that were fledged. We did not detect any differences in the performance of the 26 pairs on the low-quality dry seed-only diet against the 26 pairs that had the same dry seed, with the daily provision of the high-quality diet as well (full results in SM1). Whilst the sample size here is admittedly small, the conclusion of this diet manipulation is that Zebra Finches are extremely robust and quite capable of reproducing effectively on a very basic and nutritionally limited diet of dry seed (and they had been maintained on just the dry seed diet for over 6 weeks before any eggs were laid). With the benefit of hindsight, perhaps this is not surprising, as from an ecological perspective, the Zebra Finch lives in an ecologically poor landscape [21], and breeding is certainly known to occur from seed stored in the seedbank in the soil ([5]; Griffith pers obs), and not just in response to fresh seed produced by recent rain [23].

Our measures of reproductive success only considered immediate reproductive output in terms of both eggs and most importantly fledglings. However, the number of offspring produced may only be part of the story, and should be considered in concert with the trade-off between the quality and quantity of young (as predicted by life history theory; [32]. Studies in the laboratory have found that Zebra Finch young have the capacity, when nutritionally compromised during early development, to shift their developmental trajectory and attain a normal adult size through later compensatory growth (e.g., [33,34]). The phenomenon of flexible growth strategies by individuals, to cope with environmental, and particularly nutritional challenges early in development are a widely used adaptive strategy in animals but they can come at the expense of individual quality [35]. Given the ecological unpredictability of the Australian arid zone [21], and the high variation in food quality likely experienced by developing Zebra Finches (large quantities of nutritious green seed in good times, lower quantities of dried seed in poor times), there are interesting, and unexplored possibilities for how they can deal with this. One possibility is that, perhaps to an even greater extent than other species, Zebra Finches faced with nutritional constraints during development may minimise the immediate consequences by shifting resources to prioritise skeletal growth [35]. Whilst this may be adaptive, in enabling them to fledge quickly, and reduce the risk of predation in the nest, such strategic allocation of resources

into one aspect of development has been shown to carry long-term costs with respect to compromised cognitive ability [33], and metabolic differences in the compromised offspring when they are adults [34]. Transgenerational costs have also been observed in the Zebra Finch as a far-reaching response to nutritional constraints during rearing [36].

Furthermore, the recent study by Briga et al. [37] found complex interactions between early developmental conditions and the ability to withstand harsh conditions later in life, again highlighting the potential complexities of life history trade-offs in this species. It would be extremely useful for future work to continue exploring the complex nature of trade-offs caused by nutritional constraints in early development in the Zebra Finch, and particularly examine the variance in growth rates and the phenomenon of compensatory growth in the wild. Future work in this area will help to contextualise the work that has been performed to date. Several of the best examples of the costs of compensatory growth have come from the Zebra Finch [35], and future work in this, and other birds will hopefully be able to tell us whether Zebra Finches are extreme in this regard (due to their special ecology and diet), or whether they do indeed represent birds more generally.

Given both these delayed costs to offspring, and the parental optimisation of lifetime reproductive output (which should favour the tailoring of clutch size to the resources available), it does seem intuitive to have expected a differential investment across high- and low-quality diets. The apparent failure of adults to do so when we manipulated their diet (as described in SM1) may indicate that the dry seed diet had already exceeded a critical threshold (if they are behaving like capital breeders), and that the extra nutrients in the high-quality diet may have been superfluous and have no capacity to influence reproductive outcomes. This is in line with the earlier study by Williams [29] that found that even female Zebra Finches that received a protein supplement equivalent to 4-fold the total requirement for a five-egg clutch, over the seven-day treatment period, did not alter their clutch size. It would be good to understand what the lower nutritional threshold is for breeding in Zebra Finches and how it may differ from other similarly sized passerines. Is it that Zebra Finches are capable of squeezing more nutrients from those available in their dry seed diet compared to other species such as Great Tits that raise their offspring on a protein-rich diet of insect larvae? Alternatively, perhaps they are extracting a lower level of nutrients from their diet, but have a much lower threshold for breeding?

An impressive recent study by Pei et al. [38] meta-summarised data from fourteen years of laboratory studies of Zebra Finches and over 11,000 individuals and provides additional insight into this issue by examining the relationship between early life development and consequent reproductive success of those individuals as adults. Such an approach ignores the variation in particular traits such as cognitive ability [33], and metabolism [34], but captures their effects on the performance of the whole organism as an adult. In their study, Pei et al. [38] used nestling mass at day 8 to characterise the conditions that an individual faced during an important period of early development. Against their own intuitive expectations, they report a surprisingly low effect size for the relationship between early condition and reproductive performance of males and female in later life [38]. Indeed, they find that the proportion of variance in reproductive performance explained by the combination of inbreeding (that can be quite high in laboratory populations), age, and early condition was less than 3% in total. One of their main conclusions was that "individuals' robustness against poor conditions appears more noteworthy than their sensitivity" [38].

In conclusion, we believe that Zebra Finches are nutritionally resilient and capable of breeding (and being reared) on relatively poor-quality diets. Whilst there will inevitably be life history trade-offs to both Zebra Finch adults and offspring from the variation in nutritional environments to which they are exposed, these are likely to be rather more complex than those seen in highly seasonal breeders such as the Great Tit, where individuals only have the opportunity to breed once or twice in a lifetime during a very narrow window of opportunity. In the Zebra Finch, breeding adults, and developing offspring, appear to have a relatively high capacity to overcome nutritionally challenging conditions. As a result, whilst they are a good model system for understanding pathways to resilience in

nutritionally constrained and aseasonal, opportunistic breeding birds, they may not be a good system for understanding life history trade-offs mediated by resource availability in birds generally. The species' ability to survive and breed on such a simple, dry seed diet is one of the key characteristics that makes them so easy to keep in the laboratory. We should not lose sight of the fact that their simple diet, particularly for rearing offspring, with no supplementation required from more protein-rich food, such as insects, makes them a real outlier amongst small passerine birds. Even amongst the estrildid finches, they are apparently unique in being able to rear offspring without provisioning them with insects [5].

*1.2. Climatic Extremes during Breeding and Development*

The natural home of the Zebra Finch is the Australian arid zone covering approximately 75% of the interior of the continent [5]. The climate of this region is characterised as arid with seasonally unpredictable rainfall (spatially and temporally), in contrast to many other global deserts [21]. As a result, peaks of primary productivity can occur at any time of the year, and this is a key driver of unseasonal patterns of reproduction by many arid zone organisms in Australia [21]. In contrast, air temperatures follow a clear seasonal pattern, with hot summers and physiologically extreme air temperatures occurring from October through to February (in the Austral spring and summer). The average air temperature in which laboratory populations of Zebra Finches are held (approximately 21 °C) fairly closely reflect the average conditions during breeding in the wild, across an extended breeding period (approximately 19 °C; [39]). However, this value (19 °C) was the long-term average, and Zebra Finches breeding in the early spring will face average, minimal, and maximal conditions that are very different to those experienced by birds breeding a few months later in the summer. To contextualise the climatic conditions of breeding Zebra Finches we have characterised the daily minima and maxima air temperatures for a population of Zebra Finches and Great Tits, see Figure 1 and Supplementary Materials (SM2). The temperature range shown for the Zebra Finch here is actually fairly conservative because this distribution is based on the normal breeding period (between egg laying and the fledging of offspring) of the long-studied population at a single site—Fowlers Gap—from the first week of September to the end of January [40]. As shown by Zann [5], in other areas, Zebra Finches can breed right through winter, and further into the summer months, so both ends of the distribution are actually broader than shown. Nevertheless, even with this conservative characterisation, it is clear that the Zebra Finch breeds across a much wider range of temperatures than a typical northern hemisphere seasonally breeding species such as the well-studied Great Tit (Figure 1). This wide range of temperatures will have important consequences for breeding adults, and also developing offspring.

We should expect that the broad spectrum of climatic conditions encountered in their natural habitat will have led to the evolution of a broader thermal tolerance in the Zebra Finch [41–43]. Indeed, recent work has demonstrated that even in temperatures far exceeding their thermoneutral temperature of 30 °C, Zebra Finches can cope for sustained periods by reducing the production of metabolic heat, and increasing their intake of water and increasing evaporative cooling [44]. Zebra Finches were also revealed to have a high degree of physiological plasticity in their response to extreme heat. If water is available, they will increase consumption and evaporative cooling; if water is not available, they will tolerate a higher degree of hyperthermia [44]. Adult Zebra Finches in the wild were found to cope with air temperatures up to 45 °C for several days, with no loss of body mass, or physiological signs of stress [44]. Furthermore, Zebra Finches that had previously acclimated to high temperatures were even better able to shift their physiological response to an experimental exposure of 40 °C [45], further demonstrating the capacity of Zebra Finches to cope with extreme hot conditions, and also the potential flexibility of their thermal physiology.

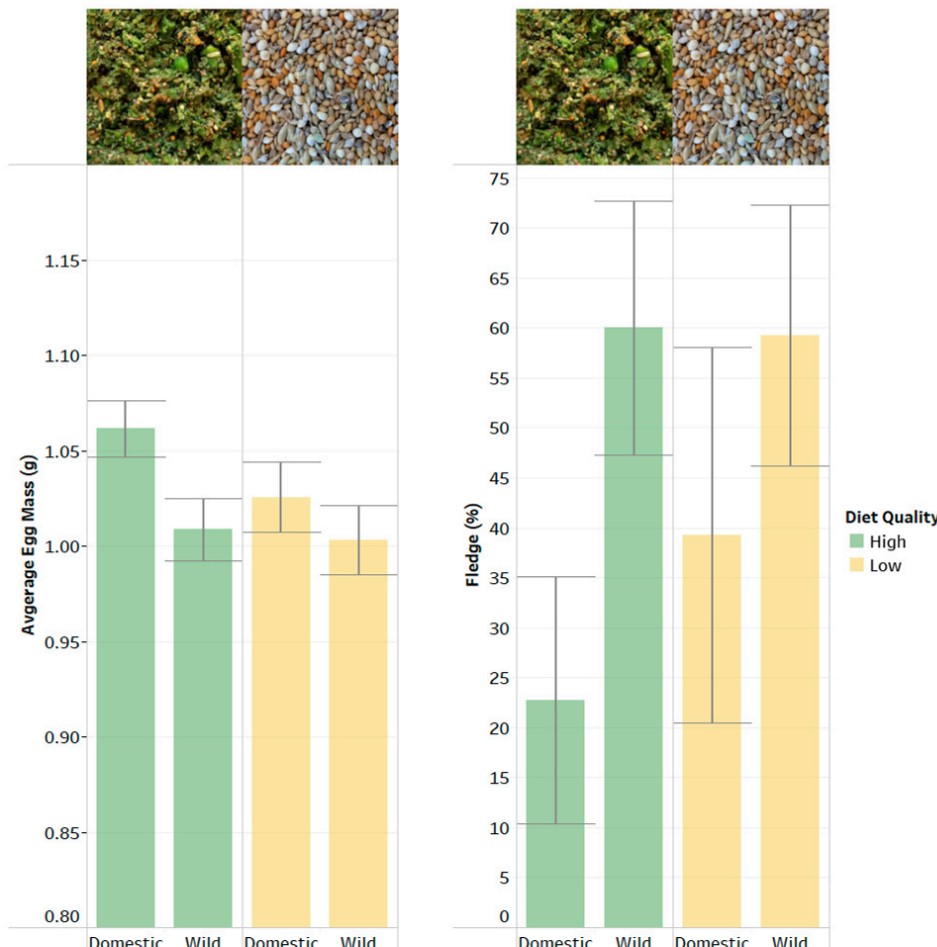

**Figure 1.** No clear effect of low- or high-quality diets on either domesticated or wild-derived Zebra Finch reproductive output in the laboratory (further details in Supplementary Material SM1).

There is an increasing focus on the ability of birds to withstand a warming and changing global climate [46,47]. The thermal tolerance of adult Zebra Finches is likely to make them a useful species for understanding the upper extremes of what might be possible, and the mechanisms, but it is important to place this in the appropriate ecological framework. The Zebra Finch is probably amongst the most heat-adapted birds in the world, and therefore might not be that useful in modelling the effects of extreme heat on less well-adapted birds in a general sense.

Whilst adult Zebra Finches may be quite able to cope with the climatic challenges of living in the Australian arid zone, reproducing during extreme conditions will be even more challenging across the extremes of the temperatures illustrated in Figure 1. For example, there are much greater energetic costs required when incubating at low temperatures, compromising both adult condition and the success of the reproductive attempt [48]. At high temperatures, the strategies used to maintain body temperature, such as sheltering in shade, increasing water consumption, and decreasing foraging activity [49,50], will be a challenge whilst investing in eggs or nestlings, again compromising either adult condition or the reproductive attempt.

Furthermore, nest temperatures have been found to exceed 40 °C regularly in the wild during active breeding [51], and variation in ambient heat across developing Zebra Finch broods has been found to influence body size [52], stress physiology [53], and the level of DNA methylation [54]. These effects may result from the effect of ambient heat exposure on either embryos or nestlings, both sensitive stages of development. At the lower end of the temperature scale, laboratory studies have also explored and demonstrated the sensitivity of Zebra Finch embryos to cold temperature [55–57]. Therefore, some Zebra

Finch offspring may start out in life developing in ambient temperatures (e.g., between 10 and 20 °C) that are very similar to those experienced by well-studied passerines in the northern hemisphere (Figure 1). Alternatively, in the same population, if they hatch from eggs laid later in the season, they may have to develop and grow in high temperatures (for example between 35 and 45 °C) that exceed their thermoneutral zone and indeed, are amongst the hottest faced by birds anywhere, and during a critical period of development. If they survive into adulthood, they will face the prospect of also dealing with this full range of conditions during their own attempts to reproduce.

An interesting area of evolutionary ecology explores the extent to which phenotypic plasticity can help individuals to address these challenges and Zebra Finches have been involved in that work. Anticipatory parental effects are where parents, based on local environmental conditions, influence offspring development so that resulting offspring phenotype is better matched to environmental conditions likely to be encountered during adulthood [58,59]. Key to this is that the conditions that are 'programmed' during offspring development are relevant to environmental conditions that will be encountered later in life [60]. There are a few compelling examples of this phenomenon occurring in some plants and insects that do not disperse far from maternal environment [61,62], but meta-analyses have found overall week effects, suggesting that anticipatory parental effects tend to be subtle or that current methodologies are not adequately capturing them [59,63]. Zebra Finches are logistically amenable to this kind of study as environmental conditions of parents and offspring can be experimentally manipulated in captivity, and there has been research on Zebra Finches into the adaptive role of parental effects resulting from nutrition [64,65] and temperature [66], as well as much discussion about the potential adaptive value of maternal effects more generally (e.g., [58,67]). However, understanding the natural history of Zebra Finches uncovers two logical flaws with the likely role of anticipatory parental effects in this species. First, Zebra Finches live for multiple years, and so each individual will have to breed across a range of conditions both within years (e.g., for temperature see Figure 1). Even if they had shorter lives, the local conditions during Zebra Finch development would offer little predictive power for the conditions that they will experience when breeding as adults, which is essential for anticipatory parental effects [60]. It was argued recently, in one of the more high profile of these studies, that in hot ambient conditions, parents signal to developing embryos to program their development to better cope with heat, and be better adapted to breeding in hot conditions as adults [66]. It is not clear what the adaptive benefit of such a strategy would be. Indeed, given the remarkable range of temperatures over which Zebra Finches breed (probably more than most other passerine birds), theory predicts that parents should be trying to maximise the range of temperatures over which offspring are physiologically prepared [68], and not trying to reduce their physiological scope by programming some offspring to be specialists in hot conditions as claimed [66].

*1.3. What Is the Life History of the Zebra Finch?*

Perhaps the most widely cited benefit of using the Zebra Finch in the laboratory is that it has a fast life history [5]. To our knowledge, no other passerine is capable of reproducing in under approximately nine months (from the end of one annual breeding season to the start of the next). Yet, the Zebra Finch is sexually mature and can breed within three months of hatching [5]. This extremely early age at sexual maturity, together with large clutches (4–9 eggs), and rapid embryonic development (12–14 days), intuitively places the Zebra Finch on the fast end of the life history continuum. These traits are also usually interpreted as evidence that the Zebra Finch is likely to be short lived. This reduced longevity appears to also fit with theoretical expectations predicting high seasonality to cause higher adult and juvenile mortality to yield populations that are held below carrying capacity, thereby minimizing food limitation compared to environments, such as the tropics, with more benign lean seasons [69]. However, all attempts to measure longevity in wild Zebra Finch are confounded by the highly mobile nature of the species, which is another

adaptation to living in such a stochastic environment [5]. Estimating the longevity of the Zebra Finch in the wild remains a key challenge. Recapture efforts such as those reported by Zann et al. [70] yield a very low estimate of longevity. However, as Zann [5] clarified, this was unlikely to reflect actual longevity, and more likely the result of birds moving outside the study area, to areas that had received more favourable rainfall. Available evidence from captive studies suggest that Zebra Finches live to approximately 4.5 years and up to 10 years [5,37], although it is very difficult to draw any conclusion from this on the likely longevity in the wild as there is no sensible way to scale from captive to wild populations [71]. Under this framework, the Zebra Finch would fit perfectly as an ideal model to draw broad inferences about north temperate songbird species that appear to share the same life history strategies, yet other aspects of their environment suggest otherwise.

The extreme level of trophic unpredictability that Zebra Finches experience in their natural habitat should select for a long/slow history. Access to food resources for the Zebra Finch, as a consequence of the spatially and temporally high variation in rainfall and prolonged droughts [21], should select for higher longevity to favour iteroparity and increase chances of surviving and reproducing in more favourable conditions [72]. Yet, higher longevity is typically, and commonly, associated with slower development and smaller clutches in the tropics [73]. This scenario raises the fundamental question of how Zebra Finches reconcile this apparent evolutionary conflict. One possible solution to explain how Zebra Finches solve this apparent evolutionary contradiction is that selection for a given clutch size and developmental rates may favour higher residual survival compared to other species (Figure 2). Increased longevity may be achieved via benefits from a relatively long period of post-natal growth and development (18–22 days), which may favour the ontogeny of DNA repair mechanisms, immune functions and cognitive abilities of higher quality [74]. Of course, extended post-natal periods in this species may also reflect low selective pressures from extrinsic mortality such as predation [75] rather than physiological trade-offs, but tests are still required to tease apart the possible causes. Studies in the southern hemisphere corroborate the increased survival hypothesis showing higher apparent survival of south temperate birds [76] and of Australian species in particular despite their high metabolism [77]. Therefore, the Zebra Finch may have relatively higher survival than that suggested by other life history traits such as early maturation, and this may set it apart from most northern hemisphere temperate species.

An alternative but not mutually exclusive possibility to the increased survival hypothesis is that Zebra Finches may instead increase annual fecundity while maintaining low survival (Figure 2). Species from arid environments have been shown to regulate the size of their clutches in response to rainfall and food availability [78]. Therefore, by increasing clutch size and undergoing repeated breeding cycles during favourable environmental conditions Zebra Finches may maximise annual offspring output and compensate for lean periods of high mortality (and no reproductive activity). This fecundity hypothesis is supported by the fact that in contrast to similar sized passerines of the northern hemisphere that are mostly constrained to breed just once a year by the seasonal availability of food [79], Zebra Finches have a much greater capacity to breed multiple times within a single calendar year and indeed over multiple years.

It takes approximately 38 days from the day the first egg is laid by a female to the fledging of their brood. In the wild, pairs will typically lay the first egg of their next attempt just over 10 days after the previous brood has fledged [80]. Therefore, in years in which they start breeding in August (i.e., 2007, Figure 3 in [40]), a pair have sufficient time to produce three successful broods of offspring before the end of the breeding season in January). With an average clutch size of approximately 5 eggs [5,40], this means that a pair of Zebra Finches can produce 15 offspring within a single breeding season. Indeed, they could achieve that before they are even 12 months old. This represents a very high level of annual fecundity for a species living in the tropics/subtropics.

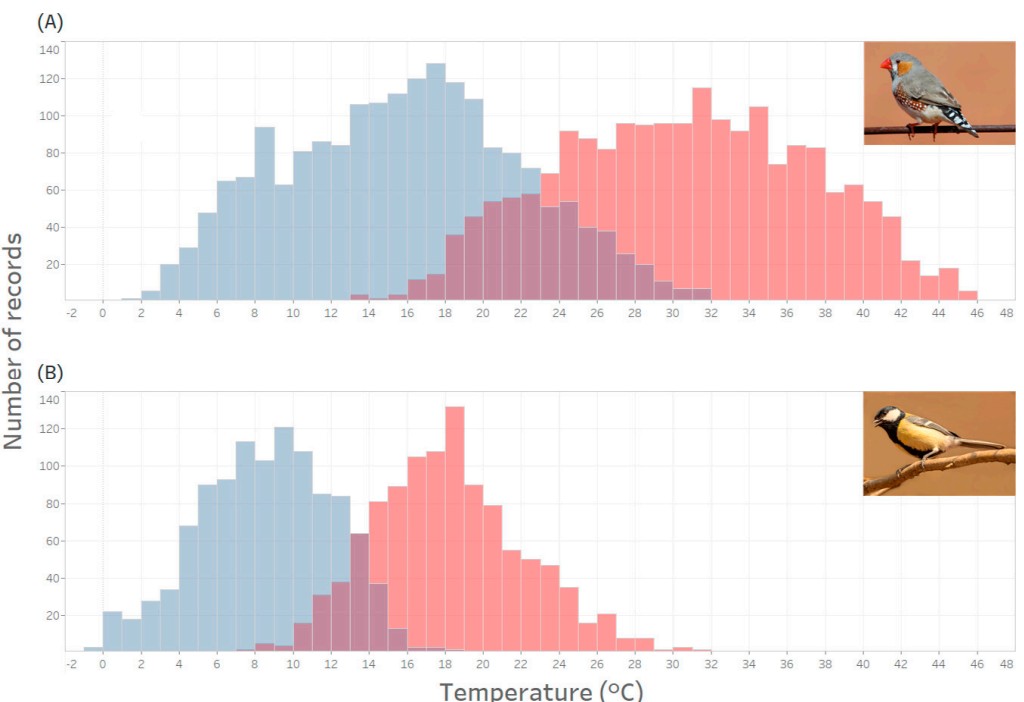

**Figure 2.** The daily minima (grey) and maxima (pink) air temperature records for the years 2005–2018 during the normal breeding period of (**A**) the Zebra Finch at Fowlers Gap, NSW, Australia, and (**B**) the Great Tit at Wytham Wood, Oxford, UK. Full details of data are given in Supplementary Material SM2. Image credits: Zebra Finch Simon Griffith; Great Tit by Petr Ganaj from Pixabay.

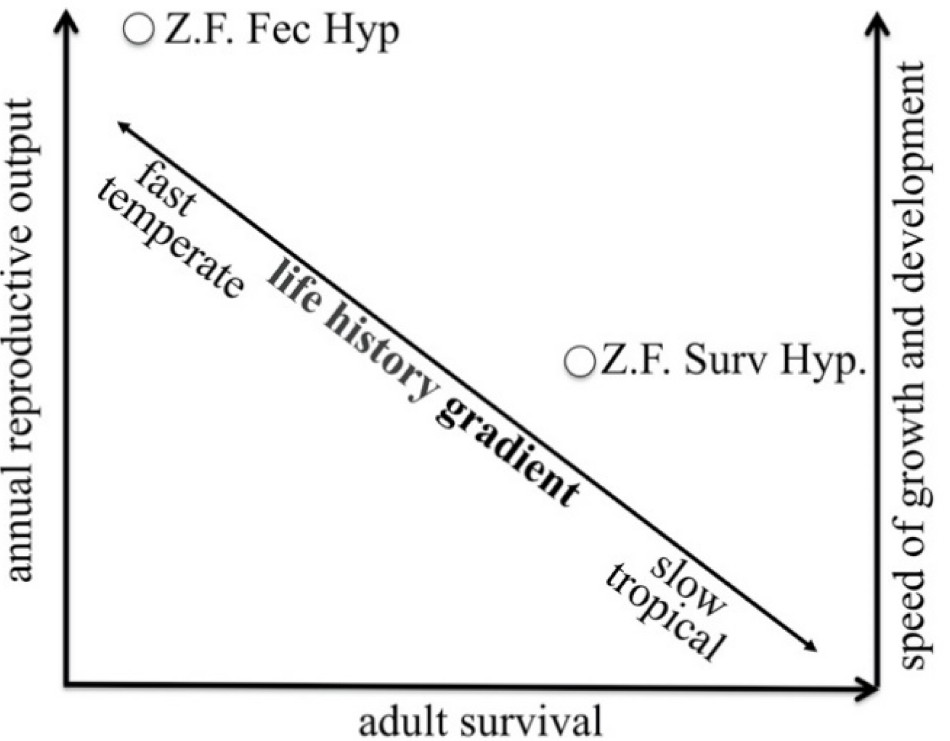

**Figure 3.** The position of the Zebra Finch in life history space. The three axes represent the phenotypic space of avian life history strategies. The diagonal arrow describes the range of possible covariation of traits along the slow–fast gradient. The two open circles reflect the combination of traits that Zebra Finches may have evolved under particular environmental conditions based on the survival and fecundity hypotheses, thus positioning them as outliers to the main axis of avian variation.

It has also been argued that the capacity to breed and multiply in quick succession has led to a number of behavioural and physiological traits in the Zebra Finch that emphasise the value of the pair bond and experience with a single partner (reviewed in [81]). In this way, the Zebra Finch is again likely to differ significantly from similarly sized passerines that tend to breed just once a year and the opportunity for re-breeding with the same partner is lower, and less valuable [81]. An example of the consequences here on research can be seen in the area of mate choice. The Zebra Finch has been the focus of dozens of studies into mate choice with rather equivocal results [82]. Typically, laboratory experiments hold Zebra Finches in single-sex groups before exposing males or females to members of the opposite sex to 'choose' a partner on the basis of a wide range of phenotypic traits. In the wild, Zebra Finches are never found in single-sex groups, and indeed are paired for life [5], spending time in very close association with their partner in breeding and non-breeding periods [83]. Given the long life history strategy of the species, and the benefits of repeatedly breeding with the same individual [81,84], it is likely that social coordination with a partner built over time is more important than mate choice on the basis of phenotypic traits that will only reliably predict quality in the short term [81]. Condition-dependent secondary sexual traits such as colour or song are likely to be of greater importance in a short-lived seasonal breeder, where pair bonds are formed immediately prior to a breeding event and the duration of bonds rarely outlasts a single reproductive event. In a species such as the Collared Flycatcher, in which males and females arrive separately from Africa onto the European breeding grounds, it makes sense for a female to base her choice of partner on whatever information she can glean from his phenotype (and territory) as quickly as possible given the short time available to breed successfully. The typical mate choice experimental approach used in research on the Zebra Finch is a very good approximation of the ecology of mate choice for a species such as the Collared Flycatcher, but does not represent the ecology of mate choice in the Zebra Finch.

Whilst we are suggesting that the Zebra Finch might fit into either a strategy or prioritising survival, or fecundity, of course it is possible that things are more complicated, with either fixed polymorphic strategies within the population, or condition-dependent plasticity within individuals. For example, with respect to sociality, it appears that, within a single population, there are a mixture of highly gregarious and colonial individuals and others that are following a more solitary existence [85]. It is possible that within a single population, some individuals follow a strategy of constrained fecundity and live longer, whilst others follow a more fast-life strategy, with greater investment in each clutch and shorter latency between broods. The inability to monitor individuals in the wild across a lifetime makes this a very difficult question to address.

Whilst many key aspects of the Zebra Finch life history are tantalisingly beyond our current knowledge, it does seem likely that the unique ecological situation of the Australian arid zone shaped strategies that made the Zebra Finch extremely easy to keep and breed in captivity, in turn making it very accessible and useful as a laboratory model. It is worth noting that the same environment in which the Zebra Finch evolved is shared by both the Budgerigar and the Cockatiel. These three species feature in internet searches for the most widely kept pet birds around the world (although a formal survey does not seem to exist to confirm this). It seems likely that these three species were shaped by this harsh and unpredictable environment, leading to the evolution of physiological, nutritional, behavioural and reproductive characteristics that enabled them to be easily kept as captive pets. Their widespread popularity is likely to be largely driven by the convenience of keeping them, their ability to be kept by non-specialists, and their capacity to breed in captivity in even the most basic contexts. All three were well established as domesticated birds by the early 20th century, and their accessibility and these same characteristics no doubt contributed to their use in scientific laboratory research. As seen in Table 1, all three species also feature in the list of the most researched non-production birds in the world.

## 2. Conclusions

Model species represent an asset for biological studies in multiple disciplines. Species that can be researched in controlled conditions, and on which an accrued research program provides a broad understanding, will remain an extremely valuable part of the 21st century biology. However, it is also important to continue to study an array of additional non-model species in an appropriate ecological context. The synthesis of knowledge from laboratory studies and a diversity of wild species will together produce a broad understanding of avian biodiversity and biology. Studies of the Zebra Finch in the laboratory make an important contribution to this effort. However, the combined effects of aseasonal breeding opportunities, the extreme climate and the ecological unpredictability typical of the Australian arid zone are likely to make the Zebra Finch quite unique amongst birds and potentially with a life history that confounds normal patterns in birds. The evolutionary strategies associated with a broad level of tolerance to environmental variation may have determined their success as the focus of so much laboratory work, but may also make the Zebra Finch somewhat of an outlier amongst birds. For the same reason, responses of Zebra Finches recorded in studies testing their physiology, behaviour, and life history may be an exception rather that the norm, i.e., the Zebra Finch may respond in one way, but most other avian species may respond quite differently. The Zebra Finch is very unlikely to represent the average, or general bird, and therefore is not a great model for capturing typical avian biology. Future studies should take these peculiarities into consideration before considering using the Zebra Finch as an ideal model species for testing proximate and ultimate responses to biotic and abiotic factors. The main issue here is not just about being careful not to use laboratory studies of a single species to generalise more broadly about all members of a particular taxon. Future studies need to think carefully about the appropriateness of using the Zebra Finch for a particular research question in the context of its evolutionary history, and behavioural and life history differences to many other passerines. Greater insight will result from future studies that interpret laboratory outcomes of future work on the Zebra Finch in the context of a better understanding of the uniqueness of this species, and its adaptations to a particular ecological context, far from the laboratories in which it will continue to be extensively used.

**Supplementary Materials:** The following are available online at https://www.mdpi.com/2673-6004/2/1/4/s1.

**Author Contributions:** All authors helped in the conception of ideas underlying the manuscript, and contributed to writing. L.L.H., S.C.G. and C.S.M. conducted the dietary manipulation presented in the supplementary material. The data analysis and preparation of the figures was done by L.L.H., R.T. and H.P.-F. All authors have read and agreed to the published version of the manuscript.

**Funding:** Please add: R.T. and L.L.H. were funded by ARC Discovery Project grants awarded to S.C.G. DP200100832 and DP170103619; C.S.M. was funded by a Macquarie University Research Excellence Scholarships; H.P.-F. was co-funded by the National Agency for Research and Development of the Republic of Chile (ANID) and an International Macquarie Research Excellence Scholarship, Scholarship award 2019—72200260.

**Institutional Review Board Statement:** The work reported in this paper was approved by the Macquarie University Animal Ethics Committee: ARA2017/54.

**Conflicts of Interest:** The authors declare no conflict of interest. The funders had no role in the design of the study; in the collection, analyses, or interpretation of data; in the writing of the manuscript, or in the decision to publish the results.

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
