# Peer review of "The Ecology of the Zebra Finch Makes It a Great Laboratory Model but an Outlier amongst Passerine Birds"

_2673-6004, doi:10.3390/birds2010004_

Round 1

Reviewer 1 Report

The zebra finch is no doubt a “great model” species for many research questions, simply because it is a “great species for laboratory work”. This statement is apparently at odds with the title of the current manuscript. After reading this manuscript, I feel convinced that the authors agree with me that the species is a great model species after all. However, the point they try to make is that the species is extremely well adapted to certain ecological conditions, which makes it less suitable as a model species for species adapted to other ecological conditions. This is rather obvious to most researchers, because no model species work this way; all species are expected to show adaptations to the environment in which they normally live. The concluding argument (p. 13): “The zebra finch is very unlikely to represent the average, or general bird, and therefore is not a great model for capturing typical avian biology” illustrates this straw man.

Having said that, I see the value of the review in summarizing the ecological adaptations that make the zebra finch a great model species for laboratory work. The paper therefore needs to shift the twist away from “not a great model” towards “why it is a great model”, and downplay the risk of using it as a model for ecological and life history adaptations for general avian biology. I would suggest emphasizing in the title how ecological adaptations makes the zebra finch a great model species for laboratory work.

Author Response

Reviewer 1

The zebra finch is no doubt a “great model” species for many research questions, simply because it is a “great species for laboratory work”. This statement is apparently at odds with the title of the current manuscript. After reading this manuscript, I feel convinced that the authors agree with me that the species is a great model species after all. However, the point they try to make is that the species is extremely well adapted to certain ecological conditions, which makes it less suitable as a model species for species adapted to other ecological conditions. This is rather obvious to most researchers, because no model species work this way; all species are expected to show adaptations to the environment in which they normally live. The concluding argument (p. 13): “The zebra finch is very unlikely to represent the average, or general bird, and therefore is not a great model for capturing typical avian biology” illustrates this straw man.

Having said that, I see the value of the review in summarizing the ecological adaptations that make the zebra finch a great model species for laboratory work. The paper therefore needs to shift the twist away from “not a great model” towards “why it is a great model”, and downplay the risk of using it as a model for ecological and life history adaptations for general avian biology. I would suggest emphasizing in the title how ecological adaptations makes the zebra finch a great model species for laboratory work.

We agree with this criticism by the reviewer of the title of the manuscript and have revised it accordingly, in order to better emphasise the adaptations to ecological conditions, that have made the zebra finch a good species for laboratory work.

The reviewer does not completely agree with the caution we are sounding over the use of the zebra finch as a model in some areas of general avian biology. However, we have refrained from adjusting the focus in the main manuscript away from that message for the reasons below. 

The reviewer suggested (in the final paragraph of their review) that we should put more emphasis on how ecological adaptations make the species a good laboratory model. We have already covered that message quite thoroughly throughout the manuscript, and yet that isn’t a particularly interesting or important message. People will use the zebra finch as it is well suited for laboratory research. Why that is, is not necessarily relevant to them.

Having now worked on the species for 15 years in both the laboratory and in the field, we feel that it is important to raise an awareness of the limitations of the zebra finch as a general passerine. The species is an outlier, and that is important to acknowledge in those areas of research where people are trying to understand birds in a general way.

As well as believing that this is an important message, it is also THE novel contribution of our manuscript, and was well received by the other two reviewers.

Reviewer 2 Report

Thanks for this well written paper, I could not agree more with!

The authors present compelling arguments why one should consider carefully for which kind of experiments the zebra finch is a sufficient model organism while emphasizing that it is still legitimate to use this birds for other, more mechanistic questions.

I propose to accept the paper as is, but would ask the authors to think about a way to make figure 2 more graphically intuitive (minor point, maybe the authors have an idea otherwise it’s ok as is)

Most of the figure 2’s content is discussed in the text, so the reason to have it is to make the information more palatable to the reader. Unfortunately the 2D representation of the mentioned 3D Information, with an additional grey level coding and double naming of both axis (x axis one inside one outside, Y axis left and right outside) makes it a bit hard to digest. I “kind of get the message”, but think it could be worked on to give a more intuitive representation of how the two proposed life history hypotheses fall into the over all picture of possible solutions.

The authors present interesting experimental data in their supplements. The journal should give them the opportunity to discuss this a bit more in the main body of the publication. If this is intentional by the authors and not owed to word count or general structure, I propose that a few strong key words could hint to the hidden results, so they can become more visible in a future literature search.

It would help the community if the authors would add a strong statement for the need of future avian bird research not only in “non standard” bird species but also for the need doing such studies in the birds' natural environment, We more and more face the situations that experiments only get approved if performed in the lab with exactly the usual four or five lab bird species and any experiment with birds from the wild or in the field get turned down as unnecessary. This could just be one sentence at the end of the conclusions to be cited in protocol applications.

Happy new year!

Author Response

Reviewer 2

Thanks for this well written paper, I could not agree more with!

The authors present compelling arguments why one should consider carefully for which kind of experiments the zebra finch is a sufficient model organism while emphasizing that it is still legitimate to use this birds for other, more mechanistic questions.

We thank the reviewer for their positive endorsement of our main message.

I propose to accept the paper as is, but would ask the authors to think about a way to make figure 2 more graphically intuitive (minor point, maybe the authors have an idea otherwise it’s ok as is)

Most of the figure 2’s content is discussed in the text, so the reason to have it is to make the information more palatable to the reader. Unfortunately the 2D representation of the mentioned 3D Information, with an additional grey level coding and double naming of both axis (x axis one inside one outside, Y axis left and right outside) makes it a bit hard to digest. I “kind of get the message”, but think it could be worked on to give a more intuitive representation of how the two proposed life history hypotheses fall into the over all picture of possible solutions.

We agree with this criticism and have amended Figure 2 to make it less cluttered, and hopefully more intuitive.

The authors present interesting experimental data in their supplements. The journal should give them the opportunity to discuss this a bit more in the main body of the publication. If this is intentional by the authors and not owed to word count or general structure, I propose that a few strong key words could hint to the hidden results, so they can become more visible in a future literature search.

We agree that the experimental data on the dietary manipulation are somewhat hidden in the paper. We felt that this data was worthy of inclusion, and it helped our general message (and the reviewer obviously agrees with this). However, we felt that we could cover this data through inclusion as supplementary material, rather than placing too much emphasis on it in the main body of the manuscript, where there may have been a danger of it detracting from the main message of the paper. We have followed the reviewers suggestion and added ‘dietary manipulation’ to the list of key words, which will help to identify this component of our work to future researchers. We have also included an additional figure in the manuscript that graphically represents the key results from the dietary manipulation. That will also help to increase the accessibility of these data for future readers.  

It would help the community if the authors would add a strong statement for the need of future avian bird research not only in “non standard” bird species but also for the need doing such studies in the birds' natural environment, We more and more face the situations that experiments only get approved if performed in the lab with exactly the usual four or five lab bird species and any experiment with birds from the wild or in the field get turned down as unnecessary. This could just be one sentence at the end of the conclusions to be cited in protocol applications.

We agree that this would be a useful inclusion and have now added the following sentences to the discussion section:

Species that can be researched in controlled conditions, and on which an accrued research program provides a broad understanding will remain an extremely valuable part of the 21st century biology. However, it is also important to continue to study an array of additional non-model species in an appropriate ecological context. The synthesis of knowledge from laboratory studies and a diversity of wild species will together produce a broad understanding of avian biodiversity and biology. Studies of the zebra finch in the laboratory make an important contribution to this effort,……

Reviewer 3 Report

Review: The zebra is a great species for laboratory work but not necessarily a great ‘model’

SC Griffith et al.

I found this manuscript very thought provoking and feel it should be required reading for all researchers using zebra finches as their model species. The MS is well organized and very well written. While the focus of the authors’ argument is primarily neurobiology, their assessment of the use of zebra finches in behavioral, reproductive biology, physiology and genomics is important especially when the species is used as a model for birds in the northern hemisphere.  I think the authors’ points could be extended into the use of the species in the field of immunology.  There are differences in the allelic frequency of MHC1 in zebra finches raised as pets when compared to wild Australian zebra finches. This may be due to the variety of pathogens the birds encounter in the field or in a colony.  However, this may be outside of the scope intended for the MS.  

Specific comments:

LN 116 “…from the only country, Australia, in which the species is found in the wild.” Are Timor zebra finches considered a separate species?

LN 368 “It has yet to be determined, but these effects…” awkward, suggest just “These effects may result form the effect of ambient heat…”

References: Despite the fact that the MS is well written, the reference section doesn’t follow the journal guidelines. It is a mixture of capitalized publication titles and abbreviated or non-abbreviated journal names.

Author Response

Reviewer 3

I found this manuscript very thought provoking and feel it should be required reading for all researchers using zebra finches as their model species. The MS is well organized and very well written. While the focus of the authors’ argument is primarily neurobiology, their assessment of the use of zebra finches in behavioral, reproductive biology, physiology and genomics is important especially when the species is used as a model for birds in the northern hemisphere.  I think the authors’ points could be extended into the use of the species in the field of immunology.  There are differences in the allelic frequency of MHC1 in zebra finches raised as pets when compared to wild Australian zebra finches. This may be due to the variety of pathogens the birds encounter in the field or in a colony.  However, this may be outside of the scope intended for the MS.  

We thank the reviewer for their positive response, and suggestion of broadening the scope of our argument by including reference to the literature on immunology. We have added the following text to the manuscript that helps to broaden the scope and include reference to these areas.

Other good examples of highly defined mechanistic questions in which the zebra finch has been adopted as a suitable model include the study of ecotoxicology and immunology. For example, recent work in the laboratory has focused on how Mercury (Hg) affects a variety of traits in the zebra finch, in an effort to understand avian responses in a general way to this common metal pollutant [15]. Similarly, in an effort to understand the likely effects of the emergent West Nile virus on birds generally, the zebra finch has been suggested as an appropriate passerine species in which laboratory trials can be conducted [16]. Even though there will be differences in the immune function of different laboratory populations [17], and indeed differences between wild and laboratory populations in the underlying architecture of immune components such the MHC [18], the zebra finch will remain one of the best species in which to conduct basic research on such highly mechanistic questions. 

Specific comments:

LN 116 “…from the only country, Australia, in which the species is found in the wild.” Are Timor zebra finches considered a separate species?

We agree that this should have been clearer. We have now amended that section as follows:

….from the only country, Australia, in which the focal subspecies is found in the wild. (Nearly all laboratory work is focussed on the Australian subspecies, which diverged from the Timor subspecies a few million years ago [Zann 1996]).

LN 368 “It has yet to be determined, but these effects…” awkward, suggest just “These effects may result form the effect of ambient heat…”

Yes we agree and have altered this as suggested.

References: Despite the fact that the MS is well written, the reference section doesn’t follow the journal guidelines. It is a mixture of capitalized publication titles and abbreviated or non-abbreviated journal names.

Yes, apologies, we had not checked the formatting of the references, but have now done so and believe that they are all compliant.

Round 2

Reviewer 1 Report

I think the authors have made a good job in revising the paper. The new title make a much better representation of the paper's main argument. The authors want to raise attention to the fact that the ZF is an ecological outlier. I am fine with that as a selling point, though I still think most researchers are already aware of the problem of generalizing ecological adaptations from a single species. It would help if the authors could come up with a few case examples to illustrate their concern on this point.
